# Know but can't Say: Exploring the Hidden Knowledge of Large Vision-Language Models for Fine-grained Perception

## Abstract

Fine-grained perception is everywhere in the real world, but it is a challenging task for Large Vision-Language Models (LVLMs), although they have shown remarkable generalization capability. How to enhance the fine-grained perception of LVLMs, achieving generalizable fine-grained perception, has become a critical research problem. In this paper, we focus on Fine-Grained Visual Classification (FGVC), a representative task of fine-grained perception. Mainstream views attribute the poor performances to the absence of relevant knowledge, such as the appearance of a specific fine-grained category, and fine-tune LVLMs with fine-grained annotated datasets. However, due to the limited scale of datasets, these approaches face the risk of overfitting, degrading the generalization capability of LVLMs. We find out that LVLMs have already been equipped with the capabilities of FGVC, which is not reflected in the generated responses. We refer to this phenomenon as *hidden knowledge*, i.e., *the model knows the answer, but cannot say it*. The existence of hidden knowledge is verified by probing techniques on LVLMs' hidden states, which reveals a gap between the internal knowledge in parameters and the external knowledge in responses. Furthermore, our probing technique discovers a generalizable, domain-invariant pattern. By leveraging this pattern, we improve the accuracy on FGVC without using annotated data of the target domain. This improvement indicates that unleashing the hidden knowledge of LVLMs can help achieve generalizable fine-grained perception.

## 1 Introduction

Fine-grained perception is fundamental to the real world: object categories can be fine-grained (Wei et al., 2021), environmental perception can be fine-grained (Li et al., 2025; Zhang et al., 2024a), and the temporal occurrence of events can be fine-grained (Qian et al., 2024; Xu et al., 2024), etc. However, fine-grained perception shows great difficulty, even challenging for humans. Recently, Large Vision-Language Models (LVLMs) have demonstrated remarkable generalization capability on open-domain tasks. However, they lack fine-grained perception abilities, particularly for Fine-Grained Visual Classification (FGVC) (Zhang et al., 2024b; Geigle et al., 2024), a representative task for fine-grained perception. In this paper, we take FGVC as an example to discuss the fine-grained perception capability of LVLMs.

FGVC aims to classify visual objects from subordinate categories (Wei et al., 2021), e.g., species of birds or models of cars, which is a common but challenging task in real-world applications. Before the era of large models, FGVC methods typically involved training a specialized model on datasets (Wah et al., 2011; Maji et al., 2013; Krause et al., 2013) with fine-grained category annotations within a specific domain. These specialized models can only perform fine-grained recognition within constrained domains such as birds or cars, which significantly limits their applicability in real-world open-domain environments. Recently, the generalization capability of LVLMs has shown promising potential to break the domain limitations of traditional FGVC. However, directly applying LVLMs to FGVC results in notably low accuracies, with a significant performance gap compared to specialized models. Considering that FGVC requires detailed knowledge about the appearance of visual objects (Chen et al., 2019), such as the color of the forehead, beak, and belly of a particular bird species, previous studies have commonly attributed the suboptimal performance of LVLMs on FGVC to

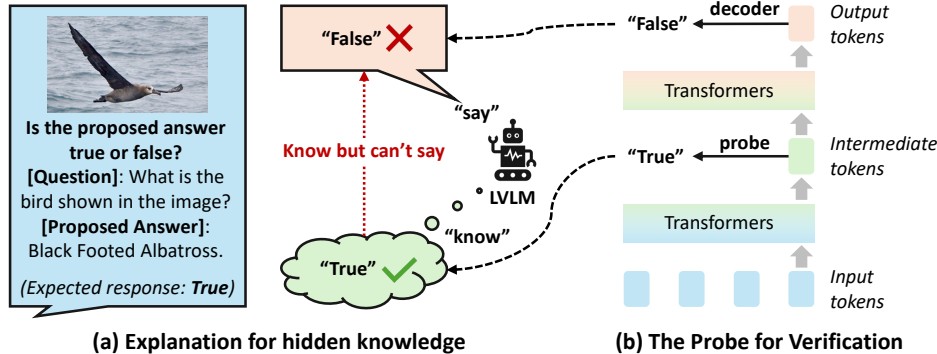

Figure 1: Examples of the hidden knowledge of LVLMs and the probe for verification. (a) Given an FGVC True/False judgment task, the model "**knows**" the answer (True) but "**can't say**" it out, leading to the wrong output (False). (b) The probe predicts what the model "**knows**" based on intermediate tokens, and the decoder predicts what the model "**says**" based on output tokens.

the lack of such knowledge. They inject such knowledge by fine-tuning LVLMs on fine-grained annotated datasets (He et al., 2025), leading to the degradation of generalization capability because fine-grained annotated data are far from sufficient to cover the open domain. Consequently, neither specialized models nor LVLMs have the capabilities of generalization and fine-grained perception simultaneously. This paper aims to address this critical issue.

We present a discovery that diverges from previous research: ***LVLMs actually have the knowledge required for FGVC, which hides behind the parameters and are not revealed in the generated responses***. As illustrated in the left part of Figure 1(a), we refer to this type of knowledge as *hidden knowledge*. The idea of hidden knowledge was motivated by an experimental comparison (Zhang et al., 2024b). Under the same setup of FGVC, the LLaVA models (Liu et al., 2024b;a) achieve accuracies of only around 10%, which is significantly lower than the 70% accuracy of their image encoder alone. This result indicates that the visual features for FGVC have already fed into the LVLMs, but they don't know how to utilize these features.

To verify the existence of hidden knowledge, as shown in Figure 1(b), we introduce the probe for verification to quantify the internal knowledge in models' parameters, and compare it with the external knowledge in generated responses. Our definitions of these two types of knowledge follow Gekhman et al. (2025). We construct a series of FGVC-based multiple-choice questions following Geigle et al. (2024). Each question is input to the LVLM together with a proposed answer, and the LVLM is instructed to output 'True' or 'False' to validate the correctness of the proposed answer. Simultaneously, applied on the last token's hidden state from an intermediate layer of the LVLM, our probe performs binary classification for the same True/False validation. Experiments across five FGVC datasets of different domains, including aircrafts (Maji et al., 2013), birds (Wah et al., 2011), cars (Krause et al., 2013), flowers (Nilsback & Zisserman, 2008), and pets (Parkhi et al., 2012), indicate that the probe's classification accuracy is consistently higher. This performance gap verifies the existence of hidden knowledge. Furthermore, we discover that the internal knowledge quantified by our probe shows cross-domain generalization capability. Under cross-domain settings where the probe is trained and tested on different domains, the internal scores are still higher than external scores from the LVLMs' responses.

After verifying the existence of hidden knowledge, we return to FGVC-based multiple-choice questions for a more comprehensive analysis of the role hidden knowledge playing in FGVC. To test the binary classification probe on multiple-choice questions, we adapt our probe to validate each option and choose the one with highest confidence. The probe based on internal knowledge still achieves higher accuracy than the model's responses. This indicates that by unleashing the hidden knowledge of LVLMs through probing techniques, it is possible to enhance fine-grained perception capabilities without fine-grained annotated data from the target domain.

Finally, we aim to the LVLMs' behavioral patterns in how they utilize the hidden knowledge. Therefore, we introduce another type of probing technique named the probe for prediction. This probe is designed to predict whether the model will generate a correct answer based on hidden states of an intermediate layer before the response is generated. Through dimensionality reduction visualizations,

we discover the separability between hidden states at intermediate layers before the model generate correct and incorrect answers. The quantitative experimental results further indicate that our probe for prediction achieves effective binary classification performance. We hope our findings provide an inspiration for future work on how to guide LVLMs to better utilize their hidden knowledge and achieve generalizable fine-grained perception.

The main contribution of this work can be summarized as: (1) We introduce a probing technique to verify the existence of hidden knowledge of LVLMs for fine-grained perception. Our probe quantifies the internal knowledge of LVLMs. Experiments show that internal scores are consistently higher than external scores on both FGVC-based True/False judgment and multiple-choice tasks. (2) Our probe shows generalization capabilities through quantitative cross-domain experiments and feature visualization on five FGVC datasets, which indicates the possibility of improving fine-grained perception capability without fine-grained annotated data of a specific domain. (3) We design another probe to capture the behavioral patterns of LVLMs on how they utilize the hidden knowledge. This probe also possesses cross-domain generalization capabilities, which helps achieve generalizable fine-grained perception.

## 2 RELATED WORKS

### 2.1 LVLMs FOR FGVC

FGVC requires distinguishing between visually similar subordinate categories, which is a challenging task for LVLMs. Based on whether other models beyond the LVLM itself are incorporated, related works can be divided into two categories in general: single-model system and multi-model system. For single-model system, He et al. (2025) tried to align the representation of visual objects and category names using attribute descriptions as an intermediate point to bind them. Despite the progress, these works fail to identify domain-specific objectives and provide justifiable explanations for their predictions. To provide justifiable explanations, Shi et al. (2025) proposed a visual rejection sampling framework, which begins by synthesizing interpretable answers that include human-verifiable visual features. After each round of fine-tuning, a reward model-free filtering mechanism is applied to select the highest-quality interpretable answers for the next round of tuning. For multi-modal system, Liu et al. (2024c) leveraged the world knowledge of LLMs as a proxy to reason about fine-grained category names and provide attributes for distinguishing subordinate categories. In this work, we aim to unleash the latent ability for FGVC lying in LVLMs themselves, without extra training data or external models.

### 2.2 HIDDEN KNOWLEDGE OF LANGUAGE MODELS

Hidden knowledge describes the phenomenon that the information required for a task is encoded in the hidden states, but the model itself generates an incorrect response, which has been discussed as a critical issue of Large Language Models. Burns et al. (2022) stepped towards discovering what models know distinct from what they say by extracting hidden knowledge in an unsupervised way. The binary classifier is optimized through Contrast-Consistent Search algorithm without the need of labels or outputs, ensuring all information is from the model's hidden knowledge. Gekhman et al. (2025) provided a formulated definition for hidden knowledge of LLMs and analyzed the hidden knowledge on factual statements, which revealed an average relative gap of 40% between internal and external knowledge. The existence of hidden knowledge explains why LLMs encode knowledge in parameters but still generate wrong answers. Zhao et al. (2024) extended the discussion of hidden knowledge to multimodal models, using the first output token to identify unanswerable questions, deceptive questions, and jailbreaking attacks through a linear probe. Mallen et al. (2023) tried to elicit hidden knowledge from LLMs fine-tuned to have specific behaviors, and demonstrated the feasibility of eliciting reliable knowledge from even untrusted LLMs. Bao et al. (2025) provided a systematic analysis of the truth direction across different LLMs and probing techniques. Experiments demonstrated that, the truth direction can be generalized to other tasks for capable models. Existing discussion on hidden knowledge has predominantly focused on LLMs, leaving visual perception tasks, especially fine-grained visual perception unexplored. This work therefore aims to investigate hidden knowledge within the domain of fine-grained visual perception.

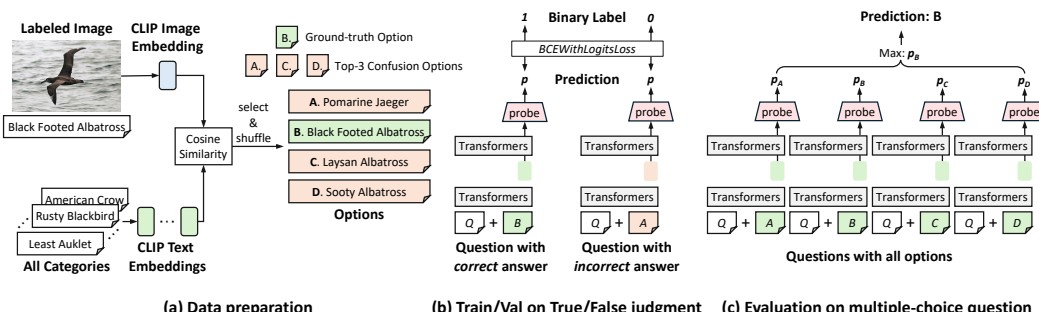

Figure 2: The pipeline of probing for verification.

# 3 METHODOLOGY

In this section, we introduce our method to verify the existence of LVLMs' hidden knowledge for FGVC and predicting LVLMs' behaviors, which mianly consists of two probing techniques: probing for verification (illustrated in Figure 1(b)) and probing for prediction (illustrated in Figure 3).

## 3.1 PRELIMINARIES

**Generation process of LVLMs.** An LVLM, denoted as $\mathcal{M}$, processes both visual and textual inputs. Given an image $\mathbf{I}$ and a text prompt $\mathbf{Q}$, $\mathcal{M}$ generates a textual response $\mathbf{A}$ in an auto-regressive way. Visual tokens $\mathbf{V}$ and text tokens $\mathbf{T}$ are concatenated to form a unified input sequence $\mathbf{X} = [\mathbf{V}; \mathbf{T}]$. This sequence is fed into a large language model denoted as $\mathcal{L}$, typically a multi-layer transformer decoder. The core of the language model $\mathcal{L}$ is a stack of $L$ transformer blocks. For an input sequence of embeddings, the $l$-th layer ($1 \leq l \leq L$) produces a sequence of hidden states $\mathbf{H}^{(l)} \in \mathbb{R}^{K \times D}$, where $K$ is the sequence length and $D$ is the hidden dimension. The hidden state for the $j$-th token at layer $l$ is denoted by $\mathbf{h}_j^{(l)}$.

In our work, we focus on the prefill stage, during which the model processes the entire input sequence $\mathbf{X}$ to compute the hidden states that will be used to generate the first output token. We are particularly interested in the hidden state of the final token of the input sequence, as this state serves as the primary contextual representation for initiating the generation process. Let this token be at position $K$. Its hidden state at layer $l$ is $\mathbf{h}_K^{(l)} \in \mathbb{R}^D$.

**Probing techniques.** Probing is an interpretability technique used to investigate what kind of information is encoded in the internal representations (i.e., hidden states) of a neural network. A probe, denoted as $\mathcal{P}$, is typically a lightweight supervised model, such as a linear classifier, a Multi-Layer Perceptron (MLP), or a Support Vector Machine (SVM). During the optimization of the probe, the parameters $\theta$ of the main model $\mathcal{M}$ are kept frozen. If the probe can successfully predict the property, it suggests the information is explicitly and accessibly encoded in the representation.

In our context, both probes for verification and prediction perform binary classification. We define a binary label $y \in \{0, 1\}$. The probes $\mathcal{P}$ take a specific hidden state $\mathbf{h}_K^{(l)}$ as input and outputs a prediction $\hat{y} \in \{0, 1\}$. The probes are trained on a dataset of tuples $\{(\mathbf{h}_{K,i}^{(l)}, y_i)\}_{i=1}^{S}$ by minimizing a classification loss. The choice of layer $l$ is a hyperparameter in our investigation.

**Evaluation of LVLMs on FGVC.** This work involves five common datasets for FGVC, including aircrafts (Maji et al., 2013), birds (Wah et al., 2011), cars (Krause et al., 2013), flowers (Nilsback & Zisserman, 2008), and pets (Parkhi et al., 2012). Following Geigle et al. (2024), we construct a series of four-choice questions, each consisting of one correct option and three confusion options with the highest CLIP (Radford et al., 2021) similarity. The expected answer includes only one letter of 'A/B/C/D'. Our experiments are conducted with mainstream LVLMs including LLaVA-1.5-7B (Liu et al., 2024b), LLaVA-NeXT-7B (Liu et al., 2024a), Qwen2.5-VL-3B, and Qwen2.5-VL-7B (Bai et al., 2025).

## 3.2 PROBING FOR VERIFICATION

The probe for verification aims to quantify the internal knowledge hidden behind LVLMs' parameters, thereby validating the existence of hidden knowledge within LVLMs through the comparison between scores of internal and external knowledge. As illustrated in Figure 1, on the FGVC-based True/False judgment task where a question and a proposed answer are input to the model, both our probe and the LVLM itself perform the same binary classification on correctness of the proposed answer. We define the binary classification accuracy of our probe as the score of internal knowledge, where the accuracy of the LVLMs' generated response is defined as the score of external knowledge. By comparing these two scores, we can verify the existence of hidden knowledge.

**Overall pipeline.** Our overall pipeline of probing for verification is illustrated in Figure 2. (1) For data preparation, we construct a series of multiple-choice questions based on FGVC datasets following Geigle et al. (2024). (2) We design a True/False judgment task, where a question together with a proposed answer is fed into the LVLM. For internal knowledge, the probe for verification is introduced to perform binary classification on whether the proposed answer is correct. The probe takes the hidden states of an intermediate layer as input, and its classification accuracy serves as the score for internal knowledge. For external knowledge, the LVLM is instructed to output 'True' or 'False' for the same binary classification task, whose accuracy is taken as the score for external knowledge. We observe that the probe's classification accuracy exceeds that of the LVLM's response, which indicates that the model's internal knowledge exceeds its external knowledge, thereby validating the existence of hidden knowledge in FGVC tasks. (3) For a deeper analysis of the role of hidden knowledge in FGVC tasks, we return to the multiple-choice task. The expected fine-grained category is selected by using the trained probe to judge the correctness of each option. Experimental results indicate that, on the multiple-choice task, our probe still achieves higher scores than the model's response. This improvement reveals the potential of unlocking hidden knowledge of LVLMs to boost their performance on fine-grained perception tasks.

**Evaluation protocol.** To quantify and compare the internal knowledge and external knowledge, we use standard binary classification metrics. Let a dataset consist of $S$ samples, with ground-truth labels $\mathbf{Y} = \{y_1, y_2, \ldots, y_S\}$. The set of predictions from the LVLM's external "True"/"False" outputs is denoted by $\hat{\mathbf{Y}}_{\text{ext}}$, while the set of predictions from our probe is denoted by $\hat{\mathbf{Y}}_{\text{int}}$. We define a scoring function, $f(\hat{\mathbf{Y}}, \mathbf{Y})$, which computes a performance score given a set of predictions and the corresponding ground-truth labels. In our work, this function $f$ represents either the Accuracy or the ROC AUC score. Then, we have $f(\hat{\mathbf{Y}}_{\text{ext}}, Y)$ as the quantification of external knowledge and $f(\hat{\mathbf{Y}}_{\text{int}}, Y)$ as that of internal knowledge.

**Prompt formulation.** For a sample from an FGVC dataset, we provide the LVLM $\mathcal{M}$ with a multimodal prompt consisting of an image $\mathbf{I}$, a question $\mathbf{Q}$, and a candidate answer $\mathbf{A}_{\text{cand}}$. This candidate answer is deliberately chosen to be either correct or incorrect, associated with a ground-truth binary label $y \in \{1, 0\}$, where $y = 1$ for a correct answer and $y = 0$ for an incorrect one. The complete prompt can be found in Appendix A.1.

**External knowledge quantification.** The LVLM's response is considered as the external knowledge. Specifically, the logits for the first generated token corresponding to the words "True" and "False" are denoted as $p(\text{'True'})$ and $p(\text{'False'})$ respectively. The model's binary classification probability $\hat{\mathbf{Y}}_{\text{ext}}$ is obtained by applying a softmax function.

**Internal Knowledge Quantification.** Simultaneously, we quantify the internal knowledge by applying a probe $\mathcal{P}_{\text{verify}}$ to the model's hidden states of the final token $\mathbf{h}_K^{(l)}$. $\mathcal{P}_{\text{verify}}$ takes a $D$-dimension vector as input, where $D$ is the dimensionality of the LVLM's hidden state. $\mathcal{P}_{\text{verify}}$ outputs a single number, which represents the probability that the proposed answer given to the LVLM is correct.

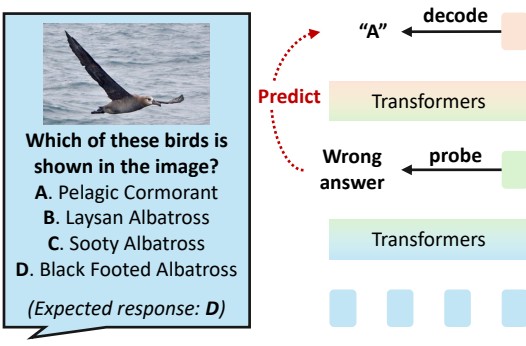

Which of these birds is shown in the image?
A. Pelagic Cormorant
B. Laysan Albatross
C. Sooty Albatross
D. Black Footed Albatross

*(Expected response: **D**)*

Figure 3: Illustration of the probe for prediction.

### Accuracy

|  |  | Aircraft | Bird | Car | Flower | Pet |
|---|---|---|---|---|---|---|
| **External** |  | 73.8 | 77.1 | 87.9 | 89.4 | 92.6 |
| **Internal** | **Aircraft** | 78.9 (+5.1) | 79.1 (+2.0) | 90.3 (+2.4) | 90.0 (+0.6) | 92.6 (0.0) |
|  | **Bird** | 76.5 (+2.7) | 79.8 (+2.7) | 90.9 (+3.0) | 90.2 (+0.8) | 93.0 (+0.4) |
|  | **Car** | 77.4 (+3.6) | 79.1 (+2.0) | 90.9 (+3.0) | 89.5 (+0.1) | 92.8 (+0.2) |
|  | **Flower** | 76.1 (+2.3) | 78.7 (+1.6) | 89.6 (+1.7) | 91.1 (+1.7) | 92.8 (+0.2) |
|  | **Pet** | 75.4 (+1.6) | 79.0 (+1.9) | 90.0 (+2.1) | 90.1 (+0.7) | 93.5 (+0.9) |

### ROC AUC

|  |  | Aircraft | Bird | Car | Flower | Pet |
|---|---|---|---|---|---|---|
| **External** |  | 85.3 | 87.7 | 96.4 | 95.3 | 97.8 |
| **Internal** | **Aircraft** | 89.4 (+4.1) | 88.2 (+0.5) | 96.9 (+0.5) | 96.3 (+1.0) | 97.8 (0.0) |
|  | **Bird** | 86.2 (+0.9) | 89.0 (+1.3) | 96.8 (+0.4) | 96.1 (+0.8) | 98.0 (+0.2) |
|  | **Car** | 87.5 (+2.2) | 88.2 (+0.5) | 97.2 (+0.8) | 96.2 (+0.9) | 97.9 (+0.1) |
|  | **Flower** | 86.2 (+0.9) | 87.8 (+0.1) | 96.3 (-0.1) | 97.7 (+2.4) | 98.0 (+0.2) |
|  | **Pet** | 85.2 (-0.1) | 87.9 (+0.2) | 96.5 (+0.1) | 96.6 (+1.3) | 98.5 (+0.7) |

Figure 4: Quantified comparison between internal scores by the probe for verification and external scores by the model's response. For internal knowledge, orange boxes represent in-domain evaluation (probe is trained and tested on the same domain), and green boxes represent cross-domain evaluation where the probe's score surpasses the external knowledge score.

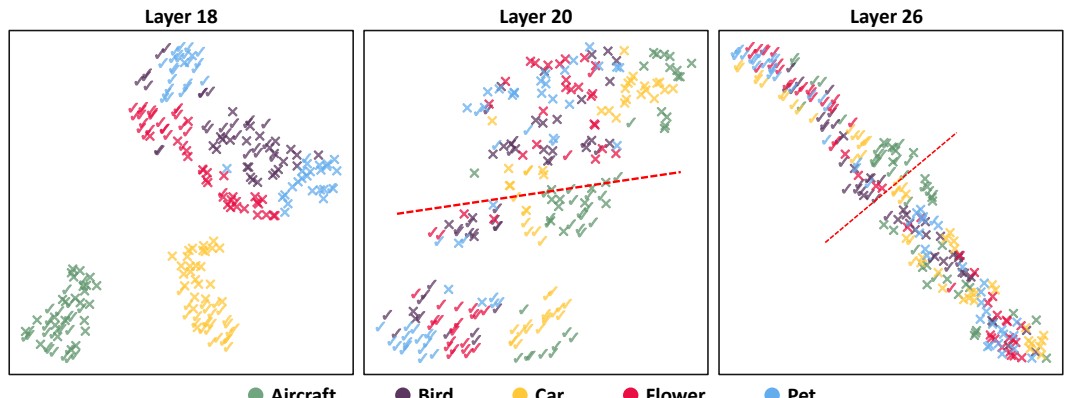

**Layer 18**  **Layer 20**  **Layer 26**

● Aircraft  ● Bird  ● Car  ● Flower  ● Pet

Figure 5: T-SNE visualization of hidden states input to the probe for verification across layers.

**Optimization of the probe.** Following the official train/test set split of the FGVC datasets, we construct the question sets for training and testing our probe according to the procedure described in Section 3.1. We then use the LVLM to answer training set questions and save the hidden states to train our probe.

### 3.3 PROBING FOR PREDICTION

After verifying the existence of hidden knowledge and its generalization capabilities, we aim to further explore LVLMs' behavioral patterns on how they utilize the hidden knowledge. Therefore, we introduce the probe for prediction as illustrated in Figure 3. Based on the hidden states of the intermediate layer, this probe predicts whether the model's output answer will be correct or incorrect before the response is generated. Under this setting, the LVLM only takes the question as input and is expected to generate the correct answer to the question.

## 4 EXPERIMENTS

### 4.1 VERIFYING THE EXISTENCE OF LVLMS' HIDDEN KNOWLEDGE

In this section, we report the experimental results of probing for verification, which reveals the existence of LVLMs' hidden knowledge through the comparison between internal knowledge and external knowledge.

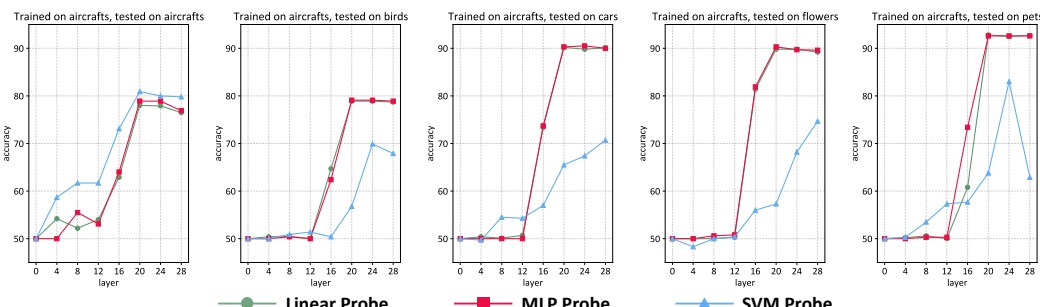

Figure 6: Comparison across probing techniques on different layers.

| Model | | LLaVA-1.5-7B | | | | | LLaVA-NeXT-7B | | | | | Qwen2.5-VL-3B | | | | |
|---|---|---|---|---|---|---|---|---|---|---|---|---|---|---|---|---|
| | | Aircraft | Bird | Car | Flower | Pet | Aircraft | Bird | Car | Flower | Pet | Aircraft | Bird | Car | Flower | Pet |
| External | | 50.0 | 50.0 | 50.0 | 50.0 | 50.0 | 50.0 | 50.0 | 50.2 | 49.9 | 50.0 | 71.6 | 74.4 | 83.9 | 84.7 | 89.0 |
| Internal | Aircraft | 60.5 | 57.1 | 67.2 | 63.2 | 68.6 | 60.2 | 56.1 | 64.9 | 59.7 | 68.0 | 76.6 | 78.4 | 87.2 | 88.4 | 92.0 |
| | Bird | 56.9 | 58.1 | 67.2 | 64.4 | 69.8 | 57.0 | 56.7 | 66.3 | 59.4 | 67.9 | 75.9 | 78.7 | 87.4 | 88.3 | 92.3 |
| | Car | 53.7 | 57.0 | 70.4 | 62.5 | 68.3 | 55.0 | 56.2 | 69.4 | 61.1 | 69.9 | 76.5 | 78.7 | 87.7 | 88.4 | 92.1 |
| | Flower | 56.3 | 57.0 | 67.4 | 66.9 | 69.0 | 52.5 | 55.4 | 64.2 | 62.8 | 69.1 | 75.3 | 77.9 | 86.5 | 88.6 | 92.1 |
| | Pet | 55.6 | 56.5 | 66.9 | 65.3 | 72.2 | 56.0 | 56.1 | 66.3 | 59.0 | 71.3 | 75.2 | 78.1 | 87.2 | 88.9 | 92.8 |
| | Avg. | 63.1 | | | | | 61.6 | | | | | 84.4 | | | | |

Figure 7: Scores for external knowledge and internal knowledge across different LVLMs, including LLaVA-1.5-7B, LLaVA-NeXT-7B, and Qwen2.5-VL-3B. Results for Qwen2.5-VL-7B are reported in Figure 4.

**Scores for internal and external knowledge** Figure 4 presents a comparison between the scores of internal knowledge and external knowledge. The external knowledge is quantified by the output response, and the internal knowledge is quantified by the binary classification of our probe as described in section 3.2.

**Cross-Domain generalization** For scores of internal knowledge, we report the performance of the probe trained and tested across different datasets. Orange cells represent where the probe is trained and tested on the same domain, and green ones indicate cross-domain settings where the internal knowledge score still surpasses the external knowledge score. Only in a few cases, marked by gray cells, does the internal knowledge score fall below that of the external knowledge. These results indicate that our probe learns features of correctness from hidden states that are more informative than the model's external response. Considering that our probe is a lightweight binary classifier with a very limited capacity, the higher score of the internal knowledge indicates that the LVLM's internal knowledge contain information that is not reflected in its response. This verifies the existence of hidden knowledge. Furthermore, the probe's cross-domain performance (the green areas) proves that it captures domain-agnostic representations. These probes have been exposed to knowledge from one domain during training, but they are still able to surpass the external knowledge score when tested on another domain, which shows the potential of hidden knowledge for generalization capabilities.

**Visualization of hidden states.** To further verify the separability of correct and incorrect inputs, we visualize the last tokens' hidden states via t-SNE technique (Maaten & Hinton, 2008). Figure 5 visualizes hidden states of Qwen2.5-VL-7B from five datasets (represented by distinct colors) with both correct (✓) and incorrect (×) answers as inputs. We can observe how the separability of the samples changes with layer depth. At layer 18, samples are still separated by domains. At

Table 1: Experimental results on multiple-choice FGVC questions under the in-domain setting, where the probes are trained and tested on the same dataset.

| Model | Method | Aircraft | Bird | Car | Flower | Pet | Average |
|---|---|---|---|---|---|---|---|
| LLaVA-v1.5-7B | External Response | 35.7 | 35.2 | 48.2 | 52.2 | 53.3 | 44.9 |
| | Probe | 54.7 | 47.4 | 65.7 | 67.2 | 71.6 | 61.3 |
| | Δ | **+19.0** | **+12.2** | **+17.5** | **+15.0** | **+18.3** | **+16.4** |
| LLaVA-NeXT-7B | External Response | 35.9 | 35.4 | 52.4 | 45.8 | 57.7 | 45.4 |
| | Probe | 56.3 | 46.6 | 65.4 | 62.7 | 71.2 | 60.4 |
| | Δ | **+20.4** | **+11.2** | **+13.0** | **+16.9** | **+13.5** | **+15.0** |
| Qwen2.5-VL-3B | External Response | 61.0 | 65.4 | 81.1 | 82.5 | 88.4 | 75.7 |
| | Probe | 67.8 | 69.0 | 83.6 | 86.8 | 90.8 | 79.6 |
| | Δ | **+6.8** | **+3.6** | **+2.5** | **+4.3** | **+2.4** | **+3.9** |
| Qwen2.5-VL-7B | External Response | 64.5 | 67.3 | 85.5 | 84.3 | 88.9 | 78.1 |
| | Probe | 73.2 | 72.2 | 88.2 | 91.3 | 93.3 | 83.6 |
| | Δ | **+8.7** | **+5.1** | **+2.7** | **+7.0** | **+4.4** | **+5.5** |

layer 20, samples of different domains get mixed together, and a consistent separation between correct and incorrect samples occurs, represented by the red dashed line. As layer 26, the boundary between samples for correct and incorrect inputs becomes more distinct. This visual evidence further validates the cross-domain generalization capability of our probe. A visualization of the complete trend with increasing layer depth can be found in Figure 11 of Appendix A.2.

**Comparison across probing techniques and layers.** As shown in Figure 6, we have tested three kinds of probes, including linear probe, MLP probe and SVM probe, on different layers. The probes are trained on aircrafts and tested on different domains. We observe that the probe fails at shallow layers, where the binary classification accuracy drops to 50%. The accuracy curve shows a sharp increase starting around the 12th layer and peaking at the 20th. This trend is consistent with our visualization in Figure 5, indicating that the most discriminative features are formed in these deeper layers. Accordingly, our subsequent experiments are mainly conducted by applying the probe to the model's deeper layers. Additionally, among different probes, while the SVM probe is superior for the in-domain setting (the first sub-figure), it generalize poorly across domains, unlike the Linear and MLP probes.

**Comparison across different LVLMs.** We evaluate the probe for verification on LVLMs including LLaVA-1.5-7B, LLaVA-NeXT-7B, Qwen2.5-VL-3B, and Qwen2.5-VL-7B. Results in Figure 7 show that stronger models (e.g., Qwen  LLaVA) reflect more internal knowledge and better cross-domain generalization. Notably, LLaVA models exhibit a strong bias towards answering 'True' to nearly any question for True/False judgment. This results in an accuracy of around 50%, indicating that they express almost no external knowledge. This phenomenon can be attributed to the "language prior" Ghatkesar et al. (2025); Liu et al. (2024d), where LVLMs are influenced by their LLM backbone, leading the decoded responses to remain invariant despite varying image and instruction inputs. This hinders the expression of internal knowledge in the model's external response. Even so, there is still detectable internal knowledge in LLaVA models with scores of around 60%.

## 4.2 UNLEASHING HIDDEN KNOWLEDGE FOR FGVC

The True/False judgment task is based on a specific input format of question-answer pairs. To validate the existence of hidden knowledge and explore its role in a more practical fine-grained perception scenario, we therefore return to the multiple-choice question task.

**Experimental setups.** On the multiple-choice task, only questions are input without proposed answers. Because our probe is a binary classifier, the multiple-choice questions are converted into a series of binary classification problems. As illustrated in Figure 2(c), we input each option into the LVLM as the proposed answer, and use the probe to judge its correctness. The option with the highest probability of correctness is selected as our probe's answer.

Table 2: Experimental results on multiple-choice FGVC questions under the **cross-domain** setting, where the probes are trained and tested on different dataset.

| Model | Method | Aircraft | Bird | Car | Flower | Pet | Average |
|-------|--------|----------|------|-----|--------|-----|---------|
| Qwen2.5-VL-3B | External Response | 61.0 | 65.4 | 81.1 | 82.5 | 88.4 | 75.7 |
|  | Probe | 62.2 | 65.9 | 82.0 | 82.2 | 88.6 | 76.2 |
|  | Δ | **+1.2** | **+0.5** | **+0.9** | **-0.3** | **+0.2** | **+0.5** |
| Qwen2.5-VL-7B | External Response | 64.5 | 67.3 | 85.5 | 84.3 | 88.9 | 78.1 |
|  | Probe | 65.1 | 67.5 | 85.0 | 85.6 | 89.1 | 78.5 |
|  | Δ | **+0.6** | **+0.2** | **-0.5** | **+1.3** | **+0.2** | **+0.4** |

**Experimental results.** Table 1 presents the comparison between the external response and our probes under the in-domain setting, i.e., the probes are trained and tested on the same dataset. Across all 4 LVLMs and all 5 datasets, the probe achieves higher multiple-choice accuracy than the models' original responses. The probe is a logistic regression model, which is applied at layer 26 for LLaVA-v1.5-7B and LLaVA-NeXT-7B, layer 30 for Qwen2.5-VL-3B, and layer 20 for Qwen2.5-VL-7B.

**Cross-domain generalization.** We also observe the cross-domain generalization capabilities of our probes under cross-domain settings, where the probes are trained and tested on different datasets. Specifically, we utilize an MLP classifier as the probe. For each target dataset, we use the combination of other four datasets to train the probe. Table 2 shows the results on Qwen2.5-VL-3B and Qwen2.5-VL-7B. Notably, our probes utilize no fine-grained supervision from the target domain, but still achieve improvements on most datasets, which verifies the cross-domain generalization capabilities of our probes.

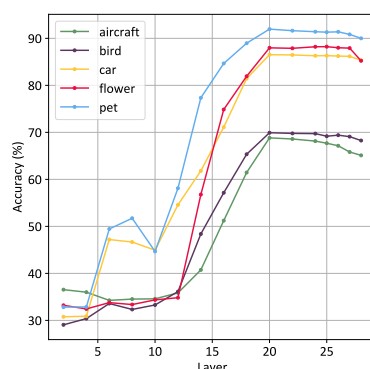

Figure 8: Accuracies on the multiple-choice task across layers of Qwen2.5-VL-7B.

**Performance across layers.** We investigate how the probe's multiple-choice accuracy varies across different layers of the LVLM, where the probe is a linear layer. As shown in Figure 8, the probe's accuracy is low in shallow layers and increases significantly in deeper layers. The transition point occurs around the 20th layer, which is consistent with our conclusions drawn from Figure 5 and Figure 6.

### 4.3 PREDICTING LVLM BEHAVIORS

**Probing for True/False Prediction** After verifying the existence of hidden knowledge and its generalization capabilities, we aim to further explore LVLMs' behavioral patterns on how they utilize the hidden knowledge. Therefore, we introduce the probe for prediction as illustrated in Figure 3. Based on the hidden states of the intermediate layer, this probe predicts whether the model's output answer will be correct or incorrect before the response is generated. Under this setting, the LVLM only takes the question as input and is expected to generate the correct answer to the question.

**Performance on binary classification.** Figure 9 presents the accuracy and ROC AUC scores of our probe trained and tested across five datasets. Orange-shaded cells represent the in-domain setting (training and testing on the same domain), while green-shaded cells represent the cross-domain setting. Most metrics are around 75% and some surpass 90%. Moreover, the consistency of the training and testing domains has a limited impact on the performance. This result indicates that the LVLM's hidden states contain domain-agnostic signals that reveal whether the model is effectively utilizing its hidden knowledge, and our probe successfully learns to capture these signals.

| Test \ Train | Aircraft | Bird | Car | Flower | Pet |
|---|---|---|---|---|---|
| Aircraft | 75.5 | 73.5 | 85.9 | 84.3 | 90.1 |
| Bird | 72.5 | 75.7 | 85.7 | 84.8 | 89.8 |
| Car | 72.8 | 72.9 | 86.7 | 84.5 | 90.2 |
| Flower | 70.5 | 72.3 | 85.5 | 88.7 | 89.3 |
| Pet | 69.3 | 72.5 | 85.6 | 84.3 | 91.6 |

**Accuracy**

| Test \ Train | Aircraft | Bird | Car | Flower | Pet |
|---|---|---|---|---|---|
| Aircraft | 83.5 | 79.4 | 84.1 | 87.4 | 89.4 |
| Bird | 81.2 | 82.7 | 83.6 | 87.4 | 88.3 |
| Car | 80.8 | 79.1 | 86.9 | 88.2 | 90.1 |
| Flower | 79.4 | 78.3 | 82.5 | 93.6 | 87.9 |
| Pet | 81.1 | 78.4 | 84.0 | 88.2 | 94.0 |

**ROC AUC**

Figure 9: Binary classification performance of the probe for prediction.

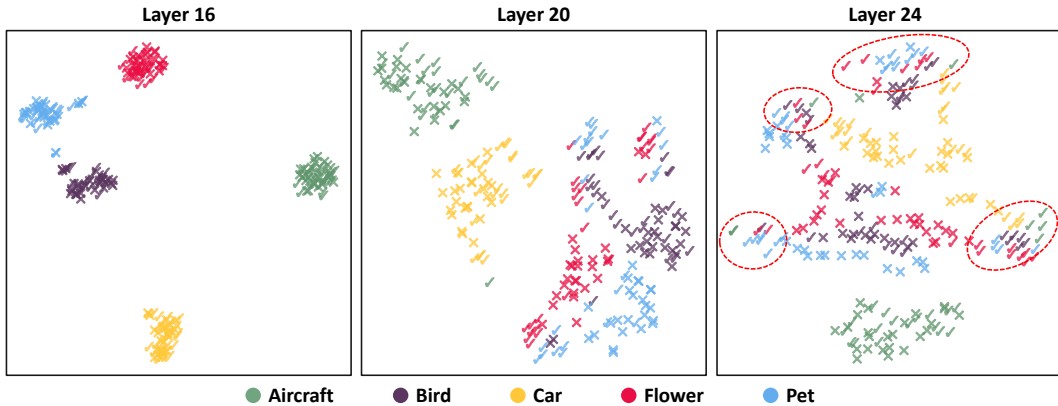

Figure 10: T-SNE visualization of hidden states input to the probe for prediction across layers.

**Visualization of hidden states.** As demonstrated in Figure 10, under this setting, the distribution of hidden states also shows a clear transformation across layers. The discrepancy between domains diminishes at deeper layers, while hidden states of correct and incorrect answers get split apart. In shallow layers, samples cluster by domain, with correct and incorrect answers mixed together. In deep layers, hidden states for correct answers (represented by ✓) cluster in groups (represented by red dashed circles), separate from those for incorrect answers. A visualization of the complete trend with increasing layer depth can be found in Figure 15 of Appendix A.2.

## 5 CONCLUSION

In this work, we have explored the hidden knowledge of LVLMs in fine-grained perception. To verify the existence of hidden knowledge, a probing technique has been designed to quantify the internal knowledge encoded in LVLMs' parameters. Quantitative experiments and visualizations have validated the existence and cross-domain generalization capabilities of hidden knowledge. Therefore, we have concluded that, the performance bottleneck of LVLMs on FGVC is not the lack of relevant knowledge. Instead, models actually *"know"* these knowledge but they *"cannot say it"*. Our probing technique helps unleash the hidden knowledge, leading to performance improvements on five FGVC datasets. Finally, we have discovered that, LVLMs' behavior on how much they utilize hidden knowledge can be predicted by hidden states, revealing the potential to guide LVLMs to better utilize their hidden knowledge and achieve generalizable fine-grained perception.

**Discussion on future work.** Our study focuses on a multiple-choice format for the FGVC task, but whether the same conclusion about hidden knowledge still hold in open-ended questions, which do not provide predefined choices? Given that a model's intermediate hidden states contain information predictive of its final answer's correctness, how this information can be leveraged to revise the output response?

## REPRODUCIBILITY STATEMENT

The LVLMs employed in this study are all publicly available, with their specific versions detailed in Section 3.1. The detailed evaluation pipeline is also described in Section 3.1. Hyper-parameters including the layers at which probes are applied and the types of probes, can be found in Section 4.2. The specific prompt templates can be found in Appendix A.1. The optimization of probes are described in Section 3.2. All experiments were conducted on NVIDIA GeForce RTX 5090 GPUs.

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

## A APPENDIX

### A.1 COMPLETE PROMPT TEMPLATE

**Prompt template for verification.** We provide the LVLM with a question from an FGVC task, together with a candidate answer that can be either correct or incorrect. The question is in a four-choice form following Geigle et al. (2024). The model is then instructed to respond with 'True' or 'False' to judge the correctness of the proposed answer. Taking a question from the aircraft dataset as an example, the complete prompt template is as follows:

```
Your job is to evaluate if a proposed answer to a fine-grained
    ↪ image classification question is correct. Here is the
    ↪ question and the proposed answer.
Question: Which of these aircrafts is shown in the image?
Choices:
A. 757-300
B. 707-320
C. 757-200
D. A310
Proposed answer: C
Is the proposed answer true or false. Just return the word 'True'
    ↪ or 'False', with no text around it.
```

**Prompt template for prediction.** Under this setting, only the question is input to the LVLM, and the model is instructed to choose one of the fine-grained category names from provided options. Taking a question from the aircraft dataset as an example, the complete prompt template is as follows:

```
Question: Which of these aircrafts is shown in the image?
Choices:
A. 757-300
B. 707-320
C. 757-200
D. A310
Answer with the letter from the given choices directly.
```

## A.2 VISUALIZATION OF HIDDEN STATES

**Hidden states input to the probe for verification.** The visualization of hidden states after every two layers of Qwen2.5-VL-7B is shown in Figure 11. In shallow layers, samples are grouped by domain (each color represents samples from one dataset, e.g., aircraft, bird, etc.). At deeper layers, starting from layer 20, samples from different domains become mixed together. Meanwhile, samples with correct and incorrect input answers begin to show a clear gap, represented by the red dashed lines. This trend indicates that, from the hidden states at deeper layers, our probe is able to learn the generalizable patterns of hidden knowledge.

**Hidden states input to the probe for prediction.** The visualization of hidden states after every two layers of Qwen2.5-VL-7B is shown in Figure 15. Similar to Figure 11, samples are grouped by domain in shallow layers. At deeper layers, starting from layer 22, samples from different domains become mixed together. Meanwhile, samples corresponding to correct predictions become clustered in certain regions, represented by the red dashed circles. This trend indicates that LVLMs' hidden states at deeper layers contain information that can predict whether the model will answer correctly, and this information is generalizable across domains.

## A.3 THE USE OF LARGE LANGUAGE MODELS

During the writing process of this paper, LLMs were used only for language polishing and error correction. We affirm that all suggestions from these models have been strictly reviewed by the authors.

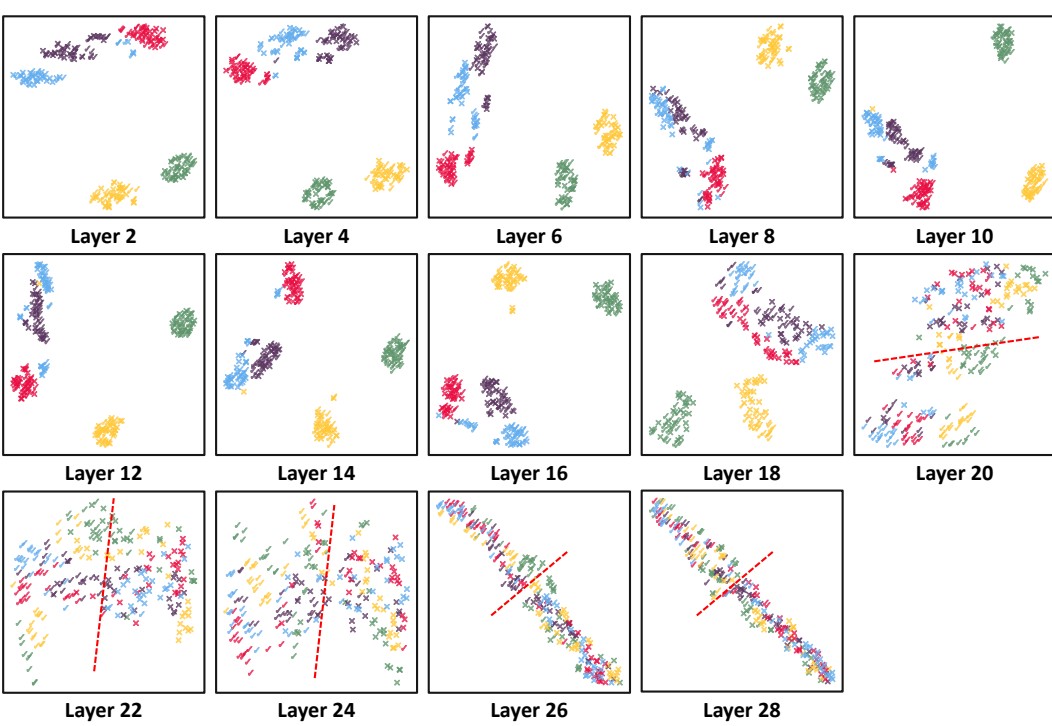

Figure 11: Visualization of the hidden states for the last token across layers. Questions and proposed answers are fed into the model, and the model is prompted to judge the correctness of the proposed answers. ✓ and × represent hidden states with correct and incorrect inputs, respectively.

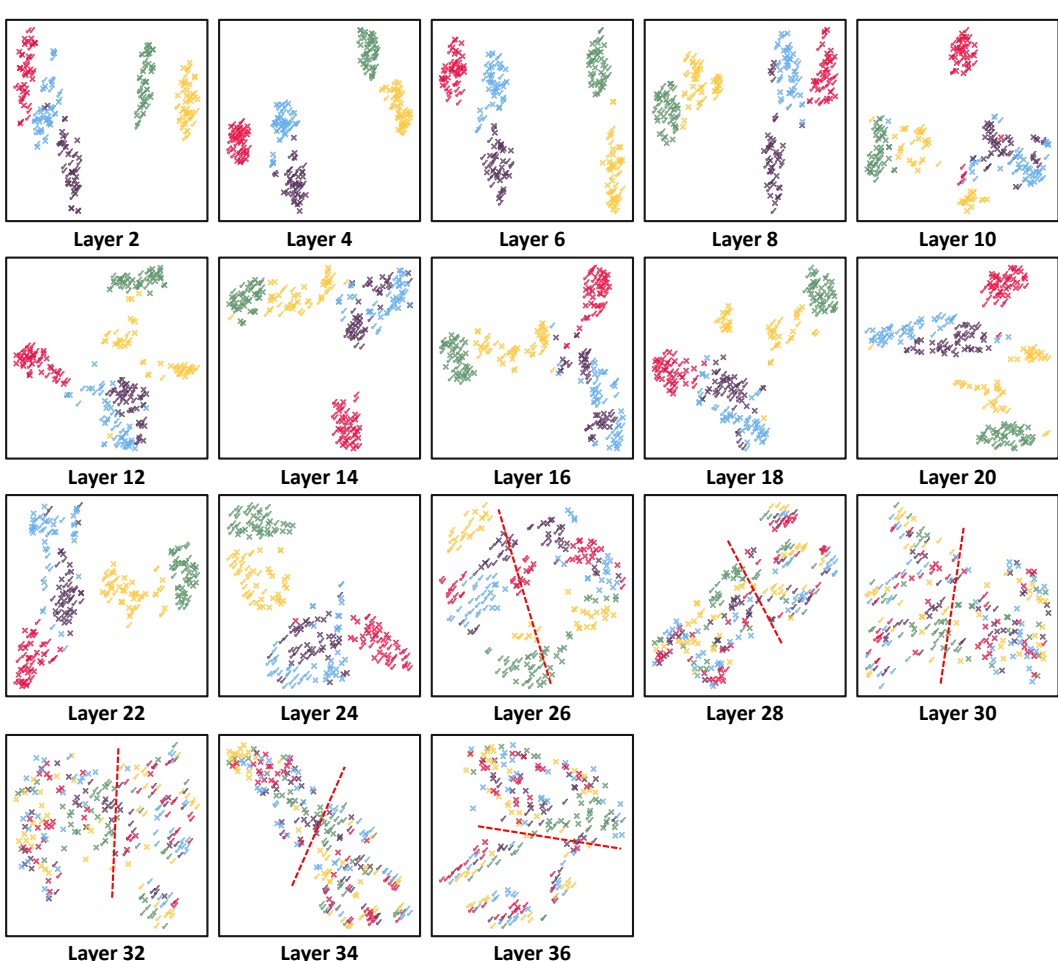

Figure 12: Visualization of the hidden states for the last token across layers of Qwen2.5-VL-3B under the same setting as Figure 11.

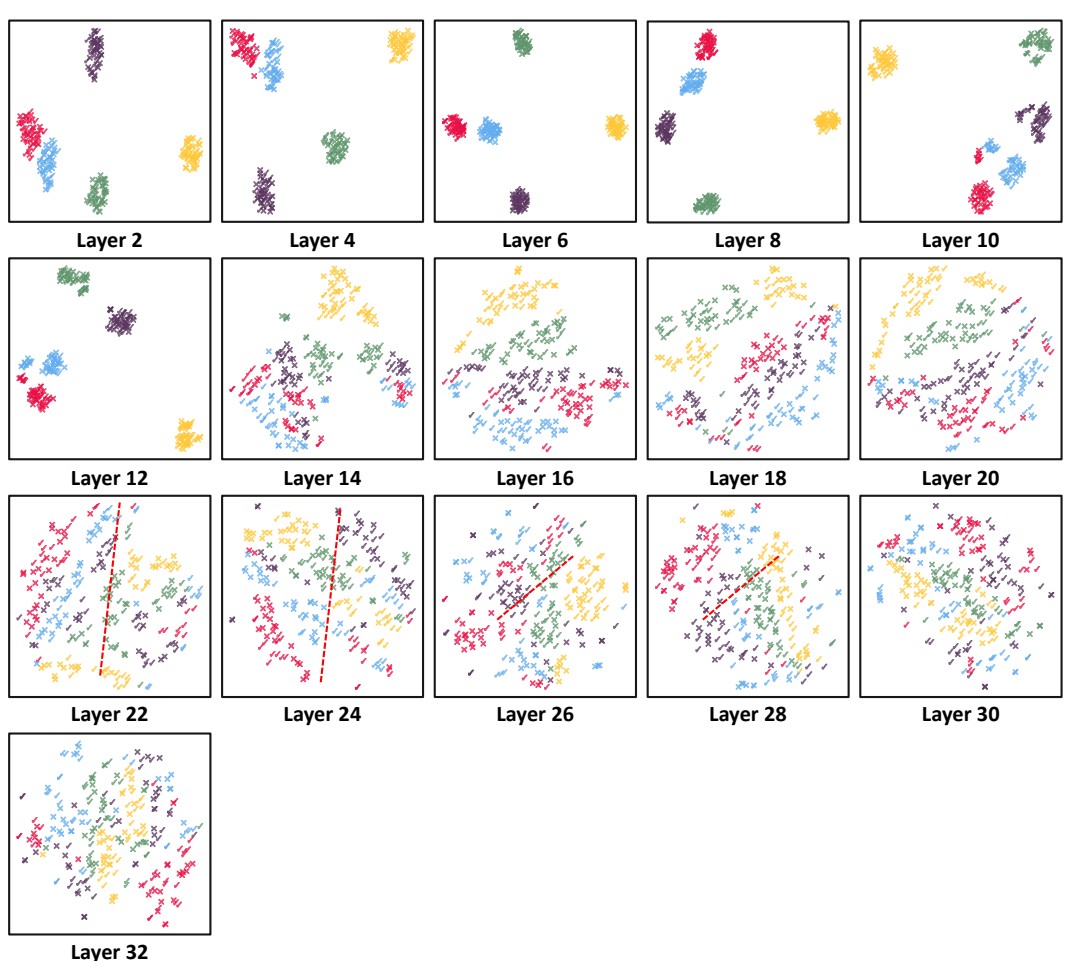

Figure 13: Visualization of the hidden states for the last token across layers of LLaVA-1.5-7B under the same setting as Figure 11.

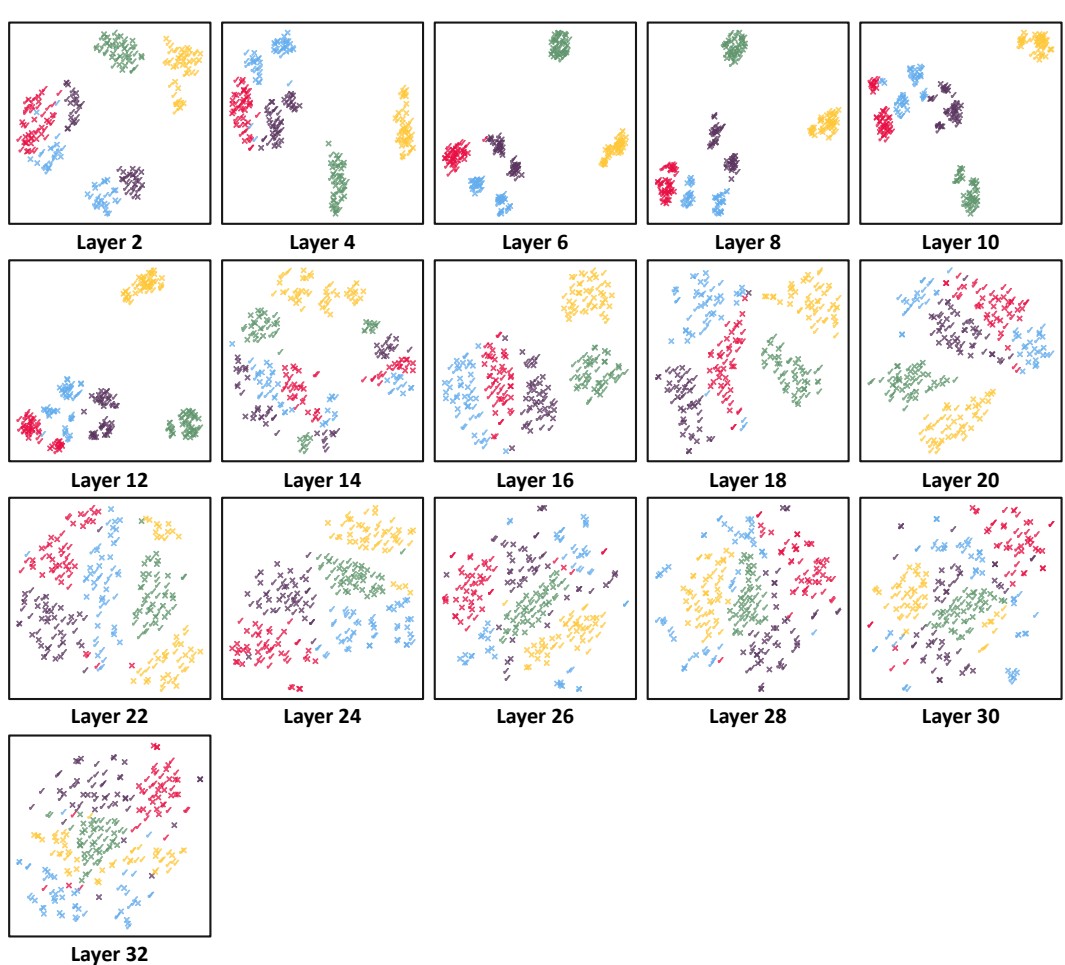

Figure 14: Visualization of the hidden states for the last token across layers of LLaVA-1.6-7B under the same setting as Figure 11.

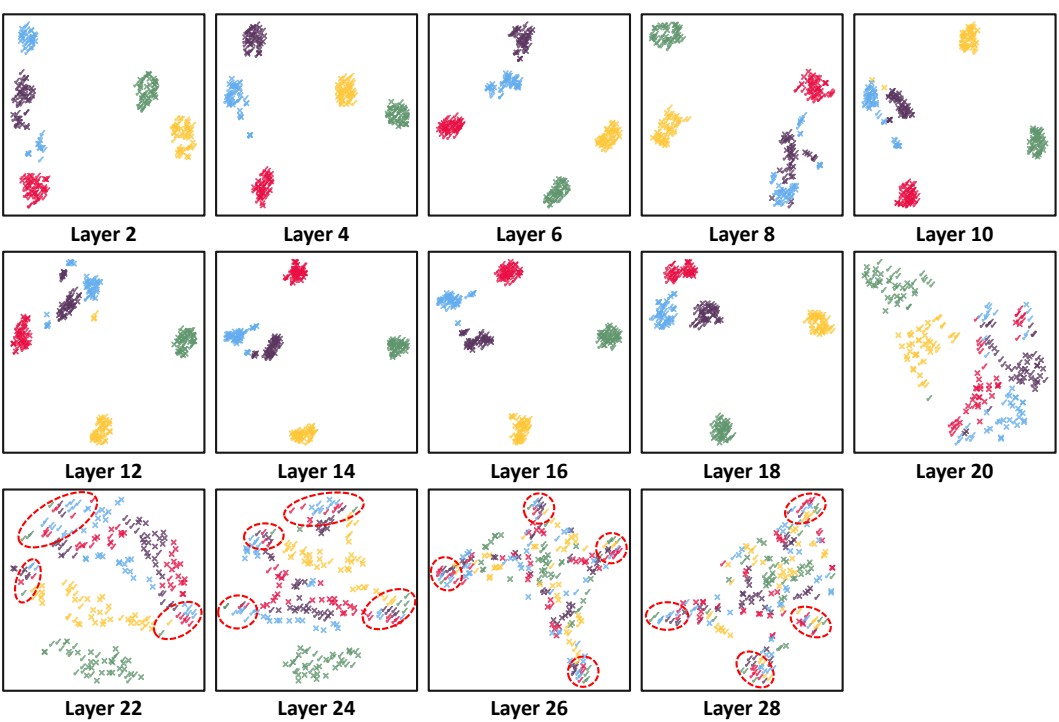

Figure 15: Visualization of the hidden states for the last token across layers. Only questions are fed into the model, and the model is prompted to choose the correct option. ✓ and × represent hidden states corresponding to correct and incorrect responses, respectively.

