# OpenReview forum: "Know but can't Say: Exploring the Hidden Knowledge of Large Vision-Language Models for Fine-grained Perception"
_ICLR.cc/2026/Conference — Submitted to ICLR 2026_

### Official Review · Reviewer_bFuN · 2025-10-17

**Soundness:** 2
**Presentation:** 2
**Contribution:** 2
**Rating:** 6
**Confidence:** 5

**Summary:**

This paper studies the task of Fine-Grained Visual Classification (FGVC) for Large Vision-Language Models (LVLMs).

The authors empirically found that LVLMs have already learned the capabilities of FGVC, yet it cannot fully leverage such knowledge to give the correct categorical responses. This exploration is interesting and can provide a good guide for future research.

Leveraging the generalizibale, domain-invariant pattern discovered from the above observation, the authors proposed techniques to improve FGVC performance of LVLMs.

**Strengths:**

- This manuscript is generally well-written.
- The experiments on verifying the existence hidden knowledge about fine-grained concepts of LVLMs are interesting and novel in the field of FGVC.
- The experiments with different probing techniques are extensive.
- The proposed technique is validated across 5 datasets and 4 LVLMs, which is sufficient.
- The proposed technique demonstrates domain generalization without using fine-grained annotations.

**Weaknesses:**

Primary weakness:
- **Lack of Comparison with highly related methods:** The most related work / method to this work are FineR [1] and Finedefics [2]. First of all, FineR [1] is training-free, zero-shot and compatible with all LVLMs, using attributes obtained from LLMs to improve FGVC of LVLMs; Thus, a comparison with FineR is necessary to fairly benchmark the performance and effectiveness of the technique proposed in this work. Second, although Finedefics [2] fine-tunes LVLMs on pseudo-labelled fine-grained data using FineR, a comparison with Finedefics is necessary under a controlled and adapted experimental setting.

- **Data Leakage Issue:** The whole study and experiments conducted in this work are on well-established, long-stand FGVC datasets. These datasets, of course, are used during the training/pre-training/fine-tuning of LVLMs and also different components inside these models (e.g., CLIP encoders). Therefore, the test data, probing data, are actually not unseen, novel data for LVLMs. This makes the observation and claim of "domain generalization without using fine-grained annotations" unclear and unsubstantiated, because the fine-grained annotations were actually used in the original LVLMs training already. The reviewer wonders, will the claims and observations of this work still hold on truly unseen/novel images contain fine-grained concepts? It is important to verify this. It does not have to be a large-scale experiments. Probably evaluation on a few hundred or a thousand pictures will suffice.

- **Vague Presentation:** To understand Figure 1, one should read the entire manuscript to get all meanings of the terms used in Figure 1 within the context of this work. This is not a a good practice. A figure appears in Intro should be intuitive and clear by itself.

[1] Liu, M., Roy, S., Li, W., Zhong, Z., Sebe, N., & Ricci, E. (2024). Democratizing fine-grained visual recognition with large language models. In ICLR, 2024.

[2] He, H., Li, G., Geng, Z., Xu, J., & Peng, Y. (2025). Analyzing and boosting the power of fine-grained visual recognition for multi-modal large language models. In ICLR 2025.

**Questions:**

Minor questions:
- The reviewer wonders, for the multiple-choice FGVC Q&As, what will happen if the condidate choices contain not only false answers under the same fine granularity, instead, containing actually "correct" answers but in coarser granularity? **For example: A. Bird / B. Black Footed Albatross / C. Albatross / D. Laysan Albatross** Will this completely change the observation?
- Please try to organize the section structure better. The current organization is hard for the readers to understand the primary contributions of this work.

---

> ### Author Response · Authors · 2025-11-24
> **Response to Reviewer bFuN**
>
> Thank you for your valuable reviews! We answer your concerns as follows:
>
> 1. **The comparison with related methods.**
>
>    - Our paper uses the same testing methodology as FineDefics [1] (4-choice questions). Furthermore, FineDefics also uses annotations from the FGVC datasets for training. Therefore, the comparison between our method and FineDefics is **fair**, and the results are presented in the table below.
>
>      |            | Aircraft | Bird     | Car      | Flower   | Pet      |
>      | ---------- | -------- | -------- | -------- | -------- | -------- |
>      | FineDefics | 63.8     | 57.6     | 84.7     | 89.9     | 92.2     |
>      | **Ours**   | **73.2** | **72.2** | **88.2** | **91.3** | **93.3** |
>
>    - However, FineR [2] utilizes custom evaluation metrics Clustering Accuracy and Semantic Similarity, which differ from those used in our work. So we cannot directly compare the results of our paper with those reported in FineR.
>
> 2. **Data Leakage Issue.**
>
>    - We agree with the reviewers that the FGVC datasets used in this paper are exposed to the pre-trained models. However, even the LVLMs that might have seen this data during their training phase still **fail to achieve accurate recognition**. In contrast, our probes are guaranteed to have had **no prior exposure** to this data (during their own training), yet they still manage to improve **cross-domain accuracy**.
>    - We are currently in the process of constructing a dataset that the model has not previously encountered, and request a few days' time.
>
> 3. **Choices with different granularities.**
>
>    Classification across different levels of granularity is an important issue in FGVC, even though most current researches focus only on categories within the same level of granularity.
>
>    We are currently investigating the behavioral patterns of LVLMs when presented with options of varying granularity. A persistent bias is observed that LVLMs **almost always select the coarse-grained option**. For instance, given three options: A. Bird, B. Albatross, and C. Black Footed Albatross, the model **consistently chooses 'Bird'**.
>
>    The goal is to **guide the model to select a finer-grained option** among a set of options that are all correct but differ in granularity. However, we have found that merely adjusting the instruction to compel the model to choose the finer-grained category is ineffective, and we are now seeking other effective solutions.
>
> 4. **Presentation and organization.**
>
>    We have revised the presentation of our paper, including the overall structure and Figure 1.
>
>    - We have removed terms such as "external knowledge" and "internal knowledge" from Figure 1. Instead, we reference the phrase "know but can't say" from the title. We have also supplemented the figure's caption.
>    - We have restructured the entire paper as follows:
>      - We have moved both probing techniques (probing for verification and probing for prediction) into a single **"3 Methodology"** section.
>      - Subsequently, all experimental results are organized into the **"4 Experiments"** section, each as a subsection corresponding to one of the paper's three main contributions: **"4.1 Verifying the Existence of LVLMs' Hidden Knowledge"** (presenting the results of probing for verification), **"4.2 Unleashing Hidden Knowledge for FGVC"** (using probes to improve the performance of multiple-choice tasks), and **"4.3 Predicting LVLM Behaviors"** (presenting the results of probing for prediction).

---

> ### Comment · Reviewer_bFuN · 2025-11-25
> **Response to Authors**
>
> Thank you for the authors for providing the rebuttal responses to address my potential concerns.
>
> The comparison with FineDefics indeed further strengthen the advantage of the proposed method. Beside, the presentation and structure of the manuscript is also improved after rebuttal. My primary concerns are well-addressed during the rebuttal.
>
> Thus, I happlily maintain my positive rating of 6 for this work.
>
> Cheers,
>
> Reviewer bFuN

---

### Official Review · Reviewer_ZPRL · 2025-10-31

**Soundness:** 2
**Presentation:** 3
**Contribution:** 3
**Rating:** 4
**Confidence:** 4

**Summary:**

This paper investigates the poor performance of Large Vision-Language Models on fine-grained visual classification (FGVC) tasks. The authors propose a "hidden knowledge" hypothesis, arguing that LVLMs do possess the necessary knowledge in their internal hidden states but fail to express it in their final output. To demonstrate this, they introduce a "verification probe" trained to extract this internal knowledge from hidden states. They show that this probe's accuracy on a True/False verification task consistently surpasses the model's direct "external" answer. Crucially, they find this internal representation of "correctness" generalizes across domains. Leveraging this discovery, the authors apply the probe to evaluate options in multiple-choice question tasks, showing that this method improves performance over the baseline model's direct answers across four LVLMs and five FGVC datasets. The paper concludes that the bottleneck for LVLMs in FGVC is a failure of knowledge extraction or alignment, not a fundamental lack of knowledge.

**Strengths:**

1. The paper's starting point (encoder 70% vs. full model 10%) is highly persuasive and clearly identifies an important problem worthy of study.
2. The paper's strongest contribution is revealing that a generalizable, cross-domain "correctness" representation exists internally in LVLMs. The cross-domain experiments in Figure 3 and the t-SNE visualizations strongly support this finding. This suggests the model's deeper layers are indeed distinguishing the abstract concept of "correct" vs. "incorrect," not just different domains.
3. The paper validates its findings across 4 different LVLMs and 5 FGVC datasets, increasing the credibility of the results.

**Weaknesses:**

1. The paper's core argument for a knowledge "gap" (Figure 3) is based on an unfair comparison. It compares the accuracy of a supervised probe (trained on "True/False" labels) with the zero-shot accuracy of the LVLM (which was not trained on this task). A supervised classifier will almost always outperform a zero-shot one. This "gap" may therefore just be an artifact of the (supervised vs. zero-shot) evaluation setup, not a true measure of "hidden" vs. "external" knowledge.
2. The paper does not explicitly state whether the probes used in Table 1 were trained in-domain or cross-domain. This is essential information for evaluating the paper's central claim. If the probe was trained in-domain (e.g., trained on "Aircraft" data to test on "Aircraft"), the entire premise of leveraging generalizable knowledge collapses, and it becomes a simple supervised method.
3. The proposed solution for MCQs requires K separate forward passes for a K-option question (one for each Question + Option_i). This results in at least a K-fold increase in inference cost compared to the baseline model. This is a very significant practical limitation that the paper completely fails to mention, making the performance gains appear computationally "free" when they are not.

**Questions:**

Please respond to the weaknesses I mentioned above.

---

> ### Author Response · Authors · 2025-11-24
> **Response to Reviewer ZPRL**
>
> Thank you for your valuable reviews! We answer your concerns as follows:
>
> 1. **The fairness of comparison between probes and the vanilla LVLM.**
>
>    We acknowledge that the comparison between probes and the vanilla LVLMs (Figures 3, Figure 6, and Table 1) is kind of unfair from the perspective of supervision. However, even though the probes are supervised, the comparison remains meaningful for the following reason:
>
>    - The results in Figures 3 and 6 indicate that the probes still **surpass the accuracy of vanilla LVLMs even under cross-dataset settings**. For example, the supervision of birds enables the probe to achieve higher accuracy on cars than the vanilla model. This demonstrates that **we do not inject knowledge specific to the target domain** (e.g., cars) into the probe but only **leverage the LVLM's own internal knowledge.**
>
>    - We perform a **fair comparison** of our probe against FineDefics [1] (ICLR 2025) under the same setting of 4-choice questions. FineDefics is also trained with fine-grained annotations. The results are presented in the table below.
>
>      |            | Aircraft | Bird     | Car      | Flower   | Pet      |
>      | ---------- | -------- | -------- | -------- | -------- | -------- |
>      | FineDefics | 63.8     | 57.6     | 84.7     | 89.9     | 92.2     |
>      | **Ours**   | **73.2** | **72.2** | **88.2** | **91.3** | **93.3** |
>
>      The results demonstrate that, under this fair comparison, our probes still achieve higher accuracy.
>
> 2. **The experimental settings of Table 1.**
>
>    The probes in Table 1 are trained in-domain. We have supplemented the cross-domain experimental results for MCQs below. In the cross-domain setting, we train the probes using a combination of the four datasets, **excluding the target dataset**. Although the improvement provided by the probes compared to the vanilla model under the cross-domain setting is relatively small, it remains significant because the training process **utilizes no fine-grained supervision from the target domain**, relying entirely on information from other domains and the LVLM's own internal knowledge. Also, the probes are optimized for binary classification tasks, with a gap in the multiple-choice tasks.
>
>    **Qwen2.5-VL-3B**:
>
>    |                       | Aircraft | Bird     | Car      | Flower   | Pet      |
>    | --------------------- | -------- | -------- | -------- | -------- | -------- |
>    | Vanilla               | 61.0     | 65.4     | 81.1     | **82.5** | 88.4     |
>    | Probes (cross-domain) | **62.2** | **65.9** | **82.0** | 82.2     | **88.6** |
>
>    **Qwen2.5-VL-7B**:
>
>    |                       | Aircraft | Bird     | Car      | Flower   | Pet      |
>    | --------------------- | -------- | -------- | -------- | -------- | -------- |
>    | Vanilla               | 64.5     | 67.3     | **85.5** | 84.3     | 88.9     |
>    | Probes (cross-domain) | **65.1** | **67.5** | 85.0     | **85.6** | **89.1** |
>
> 3. **The inference cost for MCQs.**
>
>    We agree with the reviewer that we must acknowledge that the results for the 4-choice questions in Table 1 require 4 times of forward passes, leading to an increased inference time per test sample. However, this observation does not indicate that our method introduces higher computational overhead, due to the following reasons:
>
>    - **Reduced computational cost for a single forward pass:** Our probes are applied to intermediate layers of the LLM backbone (around 3/4 of the model depth; please refer to Line 446 of the main paper), which only requires the **pre-fill phase** and **does not necessitate a complete forward pass** or the computationally expensive auto-regressive generation phase. Therefore, the computational cost of a single forward pass is lower than generating a complete response.
>    - **Fewer forward passes required:** In a large-batch testing scenario, we can first save the hidden states for each possible option. We then only need to select the hidden states corresponding to the $K$ options and subsequently execute $K$ probe predictions. As a result, in batched evaluation, the complete forward pass of the LVLM **is not required for every single test sample**.

---

### Official Review · Reviewer_X9pg · 2025-11-01

**Soundness:** 2
**Presentation:** 2
**Contribution:** 1
**Rating:** 2
**Confidence:** 3

**Summary:**

This paper investigates how to enhance fine-grained perception in Large Vision-Language Models (LVLMs), focusing on Fine-Grained Visual Classification (FGVC) as a representative task. The authors argue that LVLMs already possess fine-grained visual knowledge internally, but this knowledge is not expressed in outputs—termed “hidden knowledge” (“the model knows the answer but cannot say it”). Using probing techniques to reveal and leverage this hidden knowledge, they discover domain-invariant patterns that improve FGVC accuracy without using annotated data. The work suggests that unleashing hidden knowledge can improve the generalization ability of LVLMs on fine-grained perception tasks.

**Strengths:**

- This paper introduces a probing method to verify the existence of hidden knowledge in LVLMs related to fine-grained perception.

- Through experiments, the authors show that the intermediate hidden states contain fine-grained perceptual knowledge, but this capability is lost during response generation, as revealed by their probing method.

**Weaknesses:**

- While I agree that understanding the visual perception capability of LVLMs is highly important, the proposed probing method itself is not particularly novel or interesting. Although from a different domain, similar probing techniques have already been widely explored in the LLM community [1], along with approaches such as LogitLens [2] and PatchScope [3], which aim to interpret the hidden states of LLMs. The proposed probing method focuses on training a binary classifier using the hidden states of the LLM itself, rather than probing the vision encoder, cross-modal hidden states, or attention maps, making it conceptually similar to or derivative of these prior works.

- The terminology of “external knowledge” in the paper was somewhat confusing. Since both “internal” and “external” knowledge rely on the same intermediate representations within the LVLM, the term “external” feels somewhat inappropriate or ambiguous.

- To convincingly claim that “the model possesses internal knowledge but lacks external knowledge,” a simple classification accuracy comparison may not be sufficient. Without a more mechanistic analysis of why the model fails to generate correct answers at the final stage (e.g., from the LLM’s last layer to the LM head), it is difficult to conclude that the observed phenomenon has been thoroughly explained.

- I also find it difficult to assess whether the proposed probing method is truly reliable. In the current setup, the model is given a question and an answer and is then asked to predict “Yes” or “No.” This raises doubts about whether the trained classifier genuinely detects the presence of hidden knowledge, or whether it is simply overfitting to the Yes/No classification task.

---

References:

[1] Zhang, Anqi, et al. "Reasoning Models Know When They're Right: Probing Hidden States for Self-Verification." arXiv preprint arXiv:2504.05419 (2025).

[2] Belrose, Nora, et al. "Eliciting latent predictions from transformers with the tuned lens." arXiv preprint arXiv:2303.08112 (2023).

[3] Ghandeharioun, Asma, et al. "Patchscopes: A unifying framework for inspecting hidden representations of language models." arXiv preprint arXiv:2401.06102 (2024).

**Questions:**

See Weaknesses.

---

> ### Author Response · Authors · 2025-11-24
> **Response to Reviewer X9pg**
>
> Thank you for your valuable reviews! We answer your concerns as follows:
>
> 1. **The novelty of our work beyond similar prior works.**
>
>    We acknowledge that prior works have explored probing techniques and the interpretability of hidden states in LLMs, but the contribution of our paper is **not simply transferring existing LLM methods to LVLMs**. To the best of our knowledge, **there is no similar method applied to LVLMs previously**. Furthermore, the innovation of our paper beyond these LLM works can be summarized as follows:
>
>    - **Addressing Different Problems**: Our motivation is to achieve **generalizable fine-grained perception**, whereas most prior LLM works focus on **improving truthfulness** and **mitigating hallucination**.
>    - **Different Methodology:** FGVC is fundamentally different from the tasks addressed in LLM works (primarily TruthfulQA). Accordingly, we designed a specific pipeline as shown in Figure 2 to apply probing techniques to the FGVC task.
>    - **Different Primary Contribution:** The main contribution of our paper is to **verify the existence of hidden knowledge** in LVLMs by applying probing techniques to the FGVC task, offering a new perspective for utilizing LVLMs for fine-grained perception, **rather than designing a superior probing method**.
>
> 2. **The terminology of "external knowledge"**.
>
>    The definition of external knowledge is adopted from [1]. The score for external knowledge is calculated **based on the LVLM's response** after the final transformer layer and the decoder. Specifically, it utilizes the decoder's predicted probabilities $P(\text{'True'})$ and $P(\text{'False'})$ (Line 259 of the paper) and **does not rely on intermediate representations**. Conversely, the score for internal knowledge is calculated based on the predictions of probes applied to the intermediate representations.
>
> 3. **Analysis of the reason why LVLMs fails to generate correct answers.**
>
>    - In the main paper (Line 413), we discuss one potential reason "**language prior**" [2]: LLaVA series models almost always respond with 'True' when performing True/False judgments, no matter what the input image and instruction are, resulting in an accuracy around 50% in Figure 6. We have supplemented a discussion in the main paper explaining that this phenomenon is likely to prevent internal knowledge from being expressed in the external response.
>    - Fully explaining the reasons behind the model's failure to predict the correct answer is highly challenging and involves extensive research related to hallucination, which exceeds the scope of our paper.
>
> 4. **The overfitting issue.**
>
>    - An important finding of our paper is **the generalizability of probes across different domains** (e.g., aircraft, bird, car, etc.). As illustrated in Figure 3 of the paper, a probe trained on one dataset, when tested on other datasets, **consistently outperforms the external response**. This serves as evidence that the probe **has not overfit the training data**.
>
>    - To more thoroughly illustrate the probe's optimization, we reduce the scale of training data. Some results are shown in the table below (we have supplemented the complete experimental results in the main paper). It can be observed that, under most settings, the probe's accuracy surpasses that of the vanilla model with only **10% to 20% of the training data**.
>
>      **Qwen2.5-VL-7B**:
>
>      |  | Aircraft | Bird | Car  | Flower | Pet  |
>      |-|-|-|-|-|-|
>      | vanilla | 64.5     | 67.3 | 85.5 | 84.3   | 88.9 |
>      | 0.1     | 66.5     | 66.9 | 85.1 | 88.3   | 91.4 |
>      | 0.2     | 68.6     | 67.3 | 86.2 | 87.7   | 91.7 |
>      | 0.5     | 70.2     | 70.9 | 87.6 | 89.9   | 92.3 |
>      | 1.0     | 73.2     | 72.2 | 88.2 | 91.3   | 93.3 |
>
>      **Qwen2.5-VL-3B**:
>
>      | | Aircraft | Bird | Car  | Flower | Pet  |
>      |-|-|-|-|-|-|
>      | vanilla | 61.0 | 65.4 | 81.1 | 82.5   | 88.4 |
>      | 0.1 | 61.8 | 64.6 | 80.6 | 83.7 | 89.3 |
>      | 0.2 | 62.8 | 65.7 | 81.6 | 83.5 | 89.9 |
>      | 0.5 | 64.7 | 68.2 | 82.4 | 86.1 | 90.4 |
>      | 1.0 | 67.8 | 69.0 | 83.6 | 86.8 | 90.8 |
>
>      **LLaVA-1.5-7B**:
>
>      | | Aircraft | Bird | Car  | Flower | Pet  |
>      |-|-|-|-|-|-|
>      | vanilla | 35.7 | 35.2 | 48.2 | 52.2   | 53.3 |
>      | 0.1 | 40.8 | 35.8 | 56.0 | 53.1   | 60.8 |
>      | 0.2 | 45.9 | 40.0 | 58.8 | 52.7   | 66.0 |
>      | 0.5 | 51.1 | 43.8 | 63.8 | 62.8   | 69.7 |
>      | 1.0 | 56.7 | 47.4 | 65.7 | 67.2   | 71.6 |
>
>      **LLaVA-1.6-7B**:
>
>      |  | Aircraft | Bird | Car  | Flower | Pet  |
>      |-|-|-|-|-|-|
>      | vanilla | 35.9     | 35.4 | 52.4 | 45.8   | 57.7 |
>      | 0.1     | 44.6     | 34.4 | 52.2 | 43.0   | 58.0 |
>      | 0.2     | 49.9     | 37.2 | 58.7 | 46.3   | 61.9 |
>      | 0.5     | 53.6     | 42.2 | 62.9 | 59.9   | 68.9 |
>      | 1.0     | 56.3     | 46.6 | 65.4 | 62.7   | 71.2 |
>
>
> [1] Inside-Out: Hidden Factual Knowledge in LLMs
>
> [2] Perceiving Beyond Language Priors: Enhancing Visual Comprehension and Attention in Multimodal Models

---

### Official Review · Reviewer_Kpsc · 2025-11-01

**Soundness:** 2
**Presentation:** 3
**Contribution:** 2
**Rating:** 4
**Confidence:** 3

**Summary:**

This paper investigates Fine-Grained Visual Classification (FGVC) capabilities in Large Vision-Language Models (LVLMs). The authors demonstrate that LVLMs internally possess the knowledge necessary for FGVC, a phenomenon they term hidden knowledge, but fail to express it in their external responses. Through designed experiments, they verify the existence of this hidden knowledge. Furthermore, the study introduces a probing technique that reveals a hidden, generalizable pattern. By leveraging this pattern, the LVLMs' FGVC accuracy is improved without using any annotated data from the target domain.

**Strengths:**

* The paper is well wirtten and easy to follow, as well as well-motivated.
* The finding is interesting, as the authors reveal that the VLMs "Know" but can not "Say". The verification of the existence of hidden knowledge bring new insghts for the improvements of LVLMs not only for FGCV.
* The experiments are extensive and solid, across different model architecture, from LLaVA to QwenVL, which validates the generalization of the proposed method.

**Weaknesses:**

* Do different tasks require using hidden states from different layers to achieve the best results? Is there a pattern that can be sought?
* Could you discuss more on the potential application of the hidden knowledge beyound of FGVC?

**Questions:**

* Do different tasks require using hidden states from different layers to achieve the best results? Is there a pattern that can be sought?
* Could you discuss more on the potential application of the hidden knowledge beyound of FGVC?

---

> ### Author Response · Authors · 2025-11-24
> **Response to Reviewer Kpsc**
>
> Thank you for your valuable reviews! We answer your concerns as follows:
>
> 1. **The pattern of selecting hidden states from different layers in LVLMs.**
>
>    - **The patterns across layers**: The two settings of our paper (verification in section 3.2 and prediction in section 4) share **similar patterns** of hidden states across different layers, as demonstrated in Figure 11 and 15 in the Appendix. For Qwen2.5-VL-7B, in the **shallow layers** (before Layer 20), samples cluster according to their **categories**, such as 'aircraft', 'bird', and 'car'. In the **deep layers** (after Layer 20), samples begin to cluster based on **correctness** (represented by $\checkmark$ and $\times$), while samples from different categories start to mix together. We have supplemented the visualization results for Qwen2.5-VL-3B, LLaVA-1.5-7B, and LLaVA-1.6-7B in the Appendix (Figures 12–14), which also reveal similar patterns across layers.
>
>    - **The specific layer for the best results**: We conduct experiments across different layers of Qwen2.5-VL-7B, and the results are presented in Figure 5 of the main paper. The results of other models are listed below, where we report the accuracy of probes trained on aircrafts and tested on all five datasets. We observe that once the layer gets deeper, **the accuracy remains at a stable level**, which indicates that **selecting a specific layer is unnecessary to achieve the best results**.
>
>      **LLaVA-1.5-7B**:
>
>      | Layer | Aircraft | Bird  | Car   | Flower | Pet   |
>      | ----- | -------- | ----- | ----- | ------ | ----- |
>      | 4     | 49.98    | 50.00 | 50.01 | 50.01  | 50.01 |
>      | 8     | 49.98    | 50.00 | 50.01 | 50.01  | 50.01 |
>      | 12    | 54.16    | 50.00 | 50.01 | 50.15  | 50.01 |
>      | 16    | 60.49    | 57.35 | 65.50 | 64.77  | 68.49 |
>      | 20    | 58.57    | 57.32 | 66.07 | 63.95  | 69.28 |
>      | 24    | 58.18    | 56.70 | 65.76 | 62.99  | 67.32 |
>      | 28    | 60.01    | 57.25 | 61.85 | 64.24  | 67.89 |
>      | 32    | 59.59    | 57.13 | 66.24 | 64.86  | 68.19 |
>
>      **LLaVA-1.6-7B**:
>
>      | Layer | Aircraft | Bird  | Car   | Flower | Pet   |
>      | ----- | -------- | ----- | ----- | ------ | ----- |
>      | 4     | 50.02    | 50.00 | 50.01 | 50.01  | 50.01 |
>      | 8     | 50.02    | 50.26 | 50.01 | 50.01  | 50.01 |
>      | 12    | 55.12    | 50.00 | 49.99 | 49.99  | 50.01 |
>      | 16    | 60.91    | 56.20 | 65.65 | 61.44  | 67.73 |
>      | 20    | 59.89    | 56.20 | 64.56 | 60.03  | 68.11 |
>      | 24    | 61.21    | 56.61 | 64.44 | 59.55  | 67.32 |
>      | 28    | 59.56    | 56.18 | 64.84 | 59.49  | 69.39 |
>      | 32    | 56.65    | 55.06 | 65.03 | 59.99  | 69.34 |
>
>      **Qwen2.5-VL-3B**:
>
>      | Layer | Aircraft | Bird  | Car   | Flower | Pet   |
>      | ----- | -------- | ----- | ----- | ------ | ----- |
>      | 4     | 54.10    | 50.00 | 50.01 | 50.01  | 50.01 |
>      | 8     | 50.02    | 50.00 | 51.09 | 52.59  | 50.48 |
>      | 12    | 50.77    | 50.00 | 51.20 | 50.01  | 51.38 |
>      | 16    | 50.50    | 50.00 | 50.49 | 50.04  | 50.01 |
>      | 20    | 61.63    | 52.36 | 60.18 | 65.56  | 75.28 |
>      | 24    | 74.62    | 76.75 | 85.08 | 87.18  | 91.55 |
>      | 28    | 77.11    | 78.13 | 86.89 | 88.34  | 92.23 |
>      | 32    | 76.06    | 78.56 | 86.81 | 88.70  | 91.88 |
>      | 36    | 76.18    | 77.99 | 86.88 | 88.26  | 92.12 |
>
> 2. **The potential application of the hidden knowledge beyond FGVC.**
>
>    The exploration of hidden knowledge has made some advances in Large Language Models (LLMs), but there is **little prior work** that investigates hidden knowledge in Large Vision-Language Models (LVLMs). In LLMs, hidden knowledge is mainly related to **factual knowledge**, such as "The capital of France is Paris." Consequently, we hypothesize that in MLLMs, hidden knowledge can similarly be utilized to enhance the model's **faithfulness**, especially concerning **multimodal factual knowledge**, such as the appearance of public figures or photographs of specific events. In fact, the original motivation of this paper is to treat **Fine-Grained Visual Classification (FGVC)** as a factual task, where "the visual object belongs to some category" is a factual statement.

---

### Author Response · Authors · 2025-12-03
**Summary of paper revision**

Dear reviewers and AC,

We thank all reviewers for their valuable comments. In response to the comments, we have revised the original submission. The main updates are summarized below:

- **The reorganization of the structure** (reviewer bFuN): we have restructured the entire paper, where all techniques are moved into the "3 Methodology" section and all experiments are in the "4 Experiments" section. Each subsection of experiments corresponds to one of the paper's three main contributions.

- **Additional improvements under cross-domain settings** (reviewer ZPRL): the original results reported in the paper are under in-domain settings where the probes are trained and tested on the same domain. We have supplemented comparison results between our probes and the vanilla model under the settings, where the probes are trained without the target domain. Additional results are reported in Table 2 of the revised version.

- **Fair comparison between related methods** (reviewer ZPRL and bFuN): we have compared our probes and the related work FineDefics (ICLR 2025) under the same setting of multiple-choices. Both our method and FineDefics utilize the fine-grained supervision. The results are listed below:
     |            | Aircraft | Bird     | Car      | Flower   | Pet      |
     | ---------- | -------- | -------- | -------- | -------- | -------- |
     | FineDefics | 63.8     | 57.6     | 84.7     | 89.9     | 92.2     |
     | **Ours**   | **73.2** | **72.2** | **88.2** | **91.3** | **93.3** |

- **Polished presentation** (reviewer X9pg, ZPRL and bFuN): in addition to reconstructing the sections, we have optimized the presentation, including removing the confusing terms in Figure 1 to make it easier to understand, and supplementing analysis of the experimental results, etc. All modifications are marked in blue in the revised version.



Best regards,

Authors of submission 4667

---

### Meta-Review · Area_Chair_y2k2 · 2026-01-06

**Summary:**

The reviewers acknowledge the importance of studying visual perception in LVLMs but raise concerns that the proposed probing method is conceptually incremental, closely resembling existing LLM probing and interpretability techniques, while offering limited novelty and unclear advantages over probing vision or cross-modal representations. The framing of “external knowledge” is considered confusing, as both “internal” and “external” knowledge rely on the same hidden representations, and the central claim that models possess internal but lack external knowledge is weakly supported, relying on simple accuracy comparisons without mechanistic analysis. The reliability of the probe itself is questioned, as the supervised Yes/No classifier is unfairly compared against zero-shot LVLM performance, potentially producing an artificial “knowledge gap,” and it is unclear whether probes are trained in-domain or cross-domain, which is critical for evaluating generalization. Practical limitations are also overlooked, including a significant K-fold increase in inference cost for multiple-choice questions. Experimentally, the paper lacks comparisons with highly relevant methods such as FineR and Finedefics, raises concerns about data leakage due to the use of well-established fine-grained datasets seen during LVLM pretraining, and does not verify claims on truly novel data.

**Reviewer Concerns:**

The authors provided a good effort in their rebuttal and at least one reviewer was fully convinced by the answers. Unfortunately for the authors this is the positive reviewer who declares to be satisfied with the answers. There are however other comments which are less successful. To me one important aspect is the unfair comparison indicated by Reviewer ZPRL and partially acknowledged by the authors in their rebuttal. Also, the improvements seems marginal and the original structure of the manuscript rather unconvincing.

**Reviewer Scores:**

I think the discussion might have improved some of the opinions of the reviewers but I doubt the overall outcome would have been different. The main problem is that all the initial scores (with one exception) were rather negative and the concerns were real. This has been acknowledged also by the authors in their rebuttal. I read carefully the rebuttal and I think that the authors have done in several parts quite well. However, some of the initial concerns are still outstanding. Overall, I think the authors have done a good job in their rebuttal but the starting point was too low to change the final decision.

---

### Decision · Program_Chairs · 2026-01-26

Reject